

# Tibia functionality and Division II female and male collegiate athletes from multiple sports

Vanessa R. Yingling, Benjamin Ferrari-Church and Ariana Strickland

Department of Kinesiology, California State University, East Bay, Hayward, CA, United States of America

## ABSTRACT

**Background**. Bone strength is developed through a combination of the size and shape (architecture) of a bone as well as the bone's material properties; and therefore, no one outcome variable can measure a positive or negative adaptation in bone. Skeletal robusticity (total area/ bone length) a measure of bones external size varies within the population and is independent of body size, but robusticity has been associated with bone strength. Athletes may have similar variability in robusticity values as the general population and thus have a wide range of bone strengths based on the robustness of their bones. Therefore, the purpose of this study was to determine if an athlete's bone strength and cortical area relative to body size was dependent on robusticity. The second aim was to determine if anthropometry or muscle function measurements were associated with bone robusticity.

**Methods**. Bone variables contributing to bone strength were measured in collegiate athletes and a reference group using peripheral quantitative computed tomography (pQCT) at the 50% tibial site. Bone functionality was assessed by plotting bone strength and cortical area vs body size (body weight x tibial length) and robustness (total area/length) vs body size. Bone strength was measured using the polar strength-strain index (SSIp). Based on the residuals from the regression, an athlete's individual functionality was determined, and two groups were formed "weaker for size" (WS) and "stronger for size" (SS). Grip strength, leg extensor strength and lower body power were also measured.

**Results**. Division II athletes exhibited a natural variation in (SSIp) relative to robusticity consistent with previous studies. Bone strength (SSIp) was dependent on the robusticity of the tibia. The bone traits that comprise bone strength (SSIp) were significantly different between the SS and WS groups, yet there were minimal differences in the anthropometric data and muscle function measures between groups. A lower percentage of athletes from ball sports were "weaker for size" (WS group) and a higher percentage of swimmers were in the WS group.

**Discussion**. A range of strength values based on robusticity occurs in athletes similar to general populations. Bones with lower robusticity (slender) were constructed with less bone tissue and had less strength. The athletes with slender bones were from all sports including track and field and ball sports but the majority were swimmers.

**Conclusions**. Athletes, even after optimal training for their sport, may have weaker bones based on robusticity. Slender bones may therefore be at a higher risk for fracture under extreme loading events but also yield benefits to some athletes (swimmers) due to their lower bone mass.

Corresponding author
Vanessa R. Yingling,
vanessa.yingling@csueastbay.edu

## INTRODUCTION

Bones must be sufficiently strong to support loading from daily activities and avoid injury. Adequate bone strength during development may not only prevent injury in the short-term but may also decrease fracture incidence later in life (*Heaney et al., 2000*). However, bone strength is a complex concept and is determined by the size and shape (architecture) of a bone as well as the bone's material properties (*Van der Meulen, Jepsen & Mikić, 2001*). Therefore, one variable cannot be used to determine the bone strength or the potential for injury; multiple outcome measures are necessary to effectively monitor bones' response to exercise.

Robusticity, a measure of external bone size, is one of many genetic and anatomical factors that limit or permit bone adaptation and thus bone strength in the skeletal system (*Pandey et al., 2009*). Skeletal robusticity reflects the biological relationship between periosteal expansion relative to longitudinal growth (*Pandey et al., 2009*) and is a heritable trait established by 2 years of age. Robusticity values vary within the population but are independent of body size. The range of robusticity values (slender (low) to robust (high)) affect the ability of bone to adjust the tissue modulus or architecture to develop a sufficiently strong bone to withstand daily activities. Slender bones (low robusticity) are constructed with significantly less bone mass than more robust bones (*Schlecht & Jepsen, 2013*; *Jepsen et al., 2011*) resulting in relatively weaker bones that may be at greater risk for fracture, specifically stress fractures (*Crossley et al., 1999*; *Jones, 2002*; *Taes et al., 2010*; *Jepsen et al., 2013*). Individuals have different bone strength dependent on the robusticity of their bones. Two people with similar body sizes can have widely varied robusticity and thus varied bone strength (*Jepsen et al., 2011*). Relatively weak bones for body size may not be a problem during activities of daily living but may be detrimental under extreme loading conditions such as those experienced by athletes during training and competition. Therefore, it is important to determine if athletes have a similar variability in robusticity values as the general population and if their bone strength is affected by robusticity in similar ways that have been found in healthy populations.

Higher bone density and larger bone size (moment of inertia) have been identified in athletes compared to sedentary controls illustrating the positive effect of physical activity on bone (*Haapasalo et al., 2000*; *Greene et al., 2012*). In fact, baseball and racquet sport athletes have long term bone strength benefits, especially those athletes starting their sport during adolescence (*Kontulainen et al., 2003*; *Warden & Roosa, 2014*; *Jackowski et al., 2014*). Yet a relatively large percentage of athletes develop stress fractures accounting for 0.7% to 20% of all sports medicine clinic injuries—an indication of either relatively weak bones or excessive loading (*Fredericson et al., 2006*). Athletes cannot be grouped as a homogenous population, both intrinsic (genetic and biological) and extrinsic (environment, nutrition, training) factors affect both performance and the incidence of injury. Risk factors for injury

include training errors, training gear (shoes and orthotics), anatomical and genetic risk factors including body size, tibial width and muscle strength.

Therefore, athletes may also have a large variation in robusticity and thus bone strength values relative to body size. The purpose of this study was to determine if an athlete's bone strength and cortical area relative to body size were dependent on robusticity and if anthropometric or muscle function differed between athletes of different robusticity values. It was hypothesized that there would be a difference in bone strength and cortical area dependent on robusticity in Division II collegiate athletes but no differences in anthropometry or muscle function would be found.

## MATERIALS & METHODS

### Participants

A total of 105 university students participated in this study including 86 student athletes and 19 non-student athletes making up the reference group. Fifty-four female athletes (24.1% African American/Black, 11.1% Latina, 31.5% White, 13.0% Asian, 20.4% Mixed Race or Unknown), 37 male athletes (21.6% African American/Black, 27.0% Latino, 24.3% White, 2.7% Asian, 2.7% Pacific Islander, 21.6% Mixed Race or Unknown) and 19 referents (8 females, 11 males) (5% African American/Black, 16% Latino, 10% White, 32% Asian, 5% Pacific Islander, 32% Mixed Race or Unknown) were used in the analysis. Participants' average age was $20.7 \pm 2.2$ (18–29) years. Female athletes were members of the track and cross country (Track, CC), volleyball, soccer and swim teams. Male athletes were members of the track and cross country (Track, CC), soccer and basketball teams. All participants provided written informed consent and all study procedures were approved by the Institutional Review Board of California State University, East Bay (CSUEB-IRB-2014-004-F). A general health and demographic survey was completed. Participants were excluded if they had a history of any diseases that might influence bone mineral density (endocrine diseases, gastrointestinal disorders, and eating disorders), smoked or were pregnant.

### Anthropometry and muscle strength

Body weight and body fat percentage were measured using the BOD POD (COSMED USA, Concord, CA, USA). Height was measured in meters using a stadiometer. Maximal grip strength was tested in a standing posture with arms at sides using a hand dynamometer (BIOPAC Systems Inc, Goleta, CA, USA). Three trials were completed with a 30 second rest between each trial for both right and left hands. A relative measure of the combined force relative to body weight was then calculated. Leg extensor strength was measured using the one repetition maximum test (1 RM) on a bilateral leg-press machine (Hammer Strength-Life Fitness, Rosemont, IL, USA). Testing did not take place after practice or a weight training session or on the same day as the vertical jump test. To perform the 1 RM, participants were instructed to place feet flat on the platform, hips width apart, toes rotated slightly outward with knees flexed to 70 degrees and then extend knees to 170 degrees. After a warm up and a familiarization period, the load was set to 90–95% of their predicted 1 RM. Following each successful lift, the weight was increased by ∼5% until the participant failed

to lift the load through the entire range of motion. Approximately 3–5 min. rest periods were allowed between each trial. A repetition was considered valid when the participant used proper form and completed the lift through the full range of motion in a controlled manner without assistance. The 1RM represents the highest weight that can be lifted one time using proper technique through the full range of motion. A vertical jump test was used to estimate lower body power. Jump height was quantified using a Vertec[TM] (JUMPUSA.com, Sunnyvale, CA, USA), a common tool for measuring vertical jump ability. The Vertec[TM] is a steel structure with horizontal vanes which are rotated out of the way by the hand to indicate the height reached. A standing reach value was measured with the participants arm overhead and both feet flat on the ground. Participants then completed three counter movement vertical jumps (CMVJ). The CMVJ maximal height was calculated as the difference between the height jumped and the standing reach height. The maximal jump height of three trials was used to calculate peak power and relative peak power.

### Sayers CMVJ peak power equation (*Sayers et al., 1999*)

$$\text{Peak Power (W)} = [51.9 * \text{CMVJ height (cm)}] + [48.9 * \text{Body mass (kg)}] - 2007.$$

### Relative peak power equation

$$\text{Relative Peak Power}(W * kg^{-1}) = \text{Peak Power}(W) / \text{Body Weight (kg)}.$$

## Bone mass, structure and distribution

Bone images were obtained for the dominant tibia (*Korhonen et al., 2012*) using peripheral quantitative computed tomography (pQCT) (XCT 2000; Stratec Medizintechnik, Pforzheim, Germany). Tibia dominance was determined by asking participants, ''Which leg is your dominant leg?'' if they responded inconclusively a follow up question of, ''Which hand do you write with?'' was asked. Tibia length was measured as the distance between the medial malleolus and medial epicondyle with the knee flexed to 90 degrees. The length measurement was repeated twice, and the average was taken.

For all participants, a scout scan was performed to locate the distal end of the tibia to determine the 25% and 50% sites of the tibia length, after which the two sites were scanned. The voxel size was set to 0.5 mm, slice thickness was 2 mm and the scanning speed was 30 mm/s. The 25% site was predominately cortical diaphyseal bone. At the 50% site both cortical bone and muscle area were measured. Slice images were analyzed using manufacturer's software (version 6.20). Regions of Interest (ROI) were identified using auto find and minimize functions of the 2000L software package, manual corrections were made using visual check as necessary. Contour mode 1 with a threshold of 710 mg/cm$^3$ defined cortical bone and to determine the strength-strain index (SSIp) a contour mode of 1 and a threshold of 480 mm/cm$^3$ was used. At both the 25% and 50%-tibia sites, cortical bone mineral density, cBMD (CRT_DEN, mg/cm$^3$), total area, T.Ar (TOT_A, mm$^2$), cortical area, Ct.Ar (CRT_A, mm$^2$), periosteal perimeter, Ps.Pm (PERI_C, mm), endocortical perimeter, Ec.Pm (ENDO_C, mm), cortical thickness, Ct.Wi

(CRT_THK_C, mm) and polar moment of inertia, J (IP_CM_W, mm$^4$) and strength-strain index, SSIp (RP_CM_W, mm$^3$) were measured. Muscle cross-sectional area (mm$^2$) was determined from the 50%-tibia site. Robusticity was determined at the 25% and 50%-tibia sites as the total area divided by bone length.

SSIp = (MI/ $D_{max}$) * (CD/ND) (*Cointry et al., 2014*)

MI: Moment of Inertia

$D_{max}$ = maximum distance of a voxel from center of gravity

CD = measured cortical density (mg/cm$^3$) mineral per unit of cortical bone volume

ND = normal physiological density (1,200 mg/cm$^3$)

All scans were acquired and analyzed by 1 of 2 technicians holding Limited Permit X-ray Technician certifications from the California Department of Public Health. The short term in vivo precision (CV%) in our laboratory for all the variables used has been assessed and estimated between 0.22% and 1.7%. All scans were checked for movement artifacts at the time of the initial scan by the technician. Manufacturer supplied hydroxyapatite phantoms for pQCT were scanned daily prior to data collection.

## Robusticity and bone strength relative to body size

Whole bone strength was estimated from pQCT images using the Strength-Strain Index (SSIp). Body size was calculated as the product of body weight (BW) and tibial bone length (Le) (BW*Le). Robusticity was calculated as the total area of the bone divided by the bone length at the 50% site (Fig. 1A). The hypothesis that variation in bone strength varies by robusticity after adjusting for body size in a population of collegiate athletes was tested. Traditionally bone mechanical function is reported relative to a measure of body size. Bone functionality was assessed for males and females separately by plotting bone strength vs body size (body weight × tibia length: BW*Le) (Fig. 1C). For males and females separately, robusticity was also regressed against body size (BW*Le) (Fig. 1B). The residuals from the regressions represent the variation within SSIp and robusticity that is not explained by body size (BW * Le). Using the residuals, SSIp was then regressed against robusticity by partial regression analysis. A slope greater than zero for the partial regression indicates that bone strength is dependent on robusticity (Fig. 1D). A partial linear regression between cortical area (Ct.Ar) and robusticity accounting for body size (BW*Le) was then completed to determine any dependence of cortical area on robusticity. A slope greater than zero for the partial regression indicates that cortical area is dependent on robusticity.

Athletes and referents were then separated into two groups based on the partial regression analysis. Although a continuum in SSIp and robusticity exists, those participants with negative residuals for both SSIp and robusticity with respect to body size (BW*Le) (bottom left quadrant of Fig. 1D) were placed in a group labeled "weaker for size" (WS). The second group (top right quadrant of Fig. 1D) was comprised of individuals with positive residuals for both SSIp and robusticity with respect to body size (BW*Le) and labeled "stronger for size" (SS). Anthropometric, muscle strength and bone trait variables from the 50% tibial site were compared between groups.

Differences between the two groups (WS and SS) were determined by unpaired *t*-tests (two-tailed) with a significance value set at 0.05. All statistical analyses (*t*-tests and

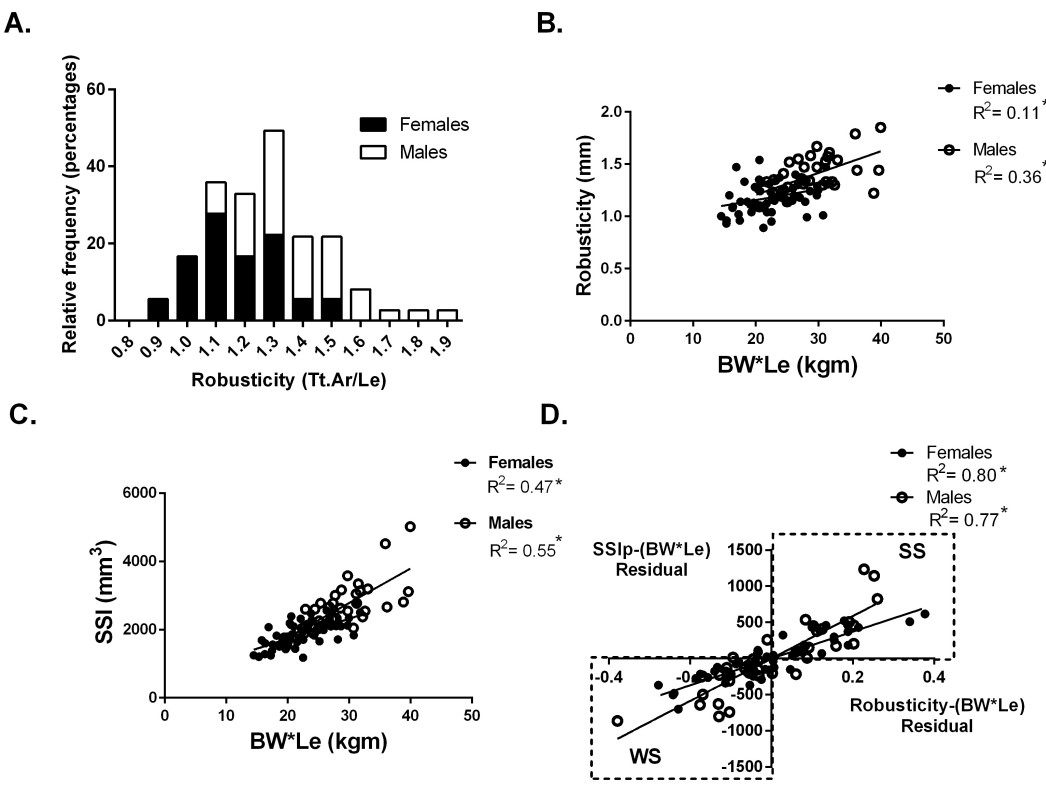

**Figure 1 Robusticity and bone strength.** (A) Tibial robusticity (Tt.Ar/Le) measured at the 50% site varied widely among females (range .89–1.54) and males (range 1.1–1.9). (B) Robusticity increased modestly with BW*Le for females ($R^2 = 0.11$) and males ($R^2 = 0.36$). (C) SSIp increased with BW*Le for females ($R^2 = 0.47$) and males ($R^2 = 0.55$). (D) Robusticity correlated significantly with SSIp for both males and females after accounting for loading (BW*Le) (females: $R^2 = 0.80$; males: $R^2 = 0.77$). * indicates significant regression $p < 0.05$.

regressions) were performed using Graph Pad (GraphPad Prism version 6.00 for Windows; GraphPad Software, San Diego, CA, USA).

## RESULTS

### Robusticity and bone strength relative to body size

At the 50% site, tibia robusticity (total cross-sectional area/tibia length) (Fig. 1A) was normally distributed with a range for females (.89–1.54) and for males (1.1–1.9). Robusticity increased modestly with body size (BW*Le), similar to previous studies (Females $R^2 = 0.11$; Males $R^2 = 0.36$) (Fig. 1B). The Strength-Strain Index (SSIp) increased with body size (BW*Le) (Females: $R^2 = 0.47$; Males: $R^2 = 0.55$) (Fig. 1C). Robusticity correlated significantly with SSIp for both males and females after accounting for body size (BW*Le) (Females: $R^2 = 0.80$; Males: $R^2 = 0.77$) (Fig. 1D), indicating that SSIp was dependent on the robusticity consistent with previous studies (*Jepsen et al., 2011*; *Jepsen et al., 2013*). The slopes of the female and male partial regression lines were significantly different. Bones

with lower robusticity values had lower SSIp levels for body size. Bone strength (SSIp) was dependent on the robusticity of the tibia (Fig. 1D).

## Anthropometrics, muscle function and bone functionality for WS vs SS groups

The tibia robusticity at the 50% site in the WS group was 17–18% less than the SS group (Figs. 2A and 2B). The WS group had SSIp values that were 27.3% and 28.8% less in females and males respectively compared to the SS group of individuals (Fig. 2C). Cortical area values were also significantly less, 19% for both females and males (Fig. 2D) in the WS groups. The largest difference between the 2 groups was found in the polar moment of inertia (J); females in the WS group had a 31.4% smaller J and males were 33.8% smaller (Fig. 2E). However, cortical bone mineral density (cBMD) was similar for both groups (Fig. 2F). Similar results for the 25% tibia site were found but not reported here.

Although the bone traits that comprise bone strength were significantly different between the SS and WS groups, there were minimal differences in the anthropometric and muscle function data between groups (Table 1). Both groups had similar heights, yet the SS group had body weights that were 8.5% greater in the females compared to the WS group. However, percent body fat was not significantly different between groups but there was a large range within groups. The female individual with the lowest body fat percentage was in the WS group and the male individual with the lowest body fat percentage was in the SS group. Tibia lengths were not significantly different between groups.

Muscle areas measured at the 50% tibia site were not different between groups for both females and males (Table 1). The three muscle strength measurements were also similar between groups. No differences were found in relative grip strength, leg extensor strength or relative power. The WS group did not lack muscle strength suggesting that differences in bone strength were not due to differences in muscular loading on the tibia.

## Type of sport for WS and SS groups

Table 2 indicates the percentage of individuals in the WS group. Similar numbers of female and male runners (Track/CC) were in the WS group, 44% and 43% respectively. A lower percentage of athletes from ball sports were "weaker for size" (WS group); 30% and 33.3% respectively for females and males. The ball sport category included volleyball and soccer for females and soccer and basketball for males. A large percentage of swimmers, 72% were in the WS group. The number of referent individuals in the WS group was 75% for females and 82% for males.

## Cortical area in WS and SS groups

Adapting a bone to optimize bone strength may be limited by the cortical area of that bone. To determine if any athletes or referent participants were "weaker for size" but fully adapted based on cortical area, a partial regression analysis of cortical area and robusticity accounting for BW*Le was done. If an individual had an expected or greater than expected cortical area for their body size (positive residual values for Ct.Ar vs body size) they may also have relatively weak bones for body size but the tibia reached the highest strength biologically possible based on their robusticity. Cortical area (Ct.Ar) was larger

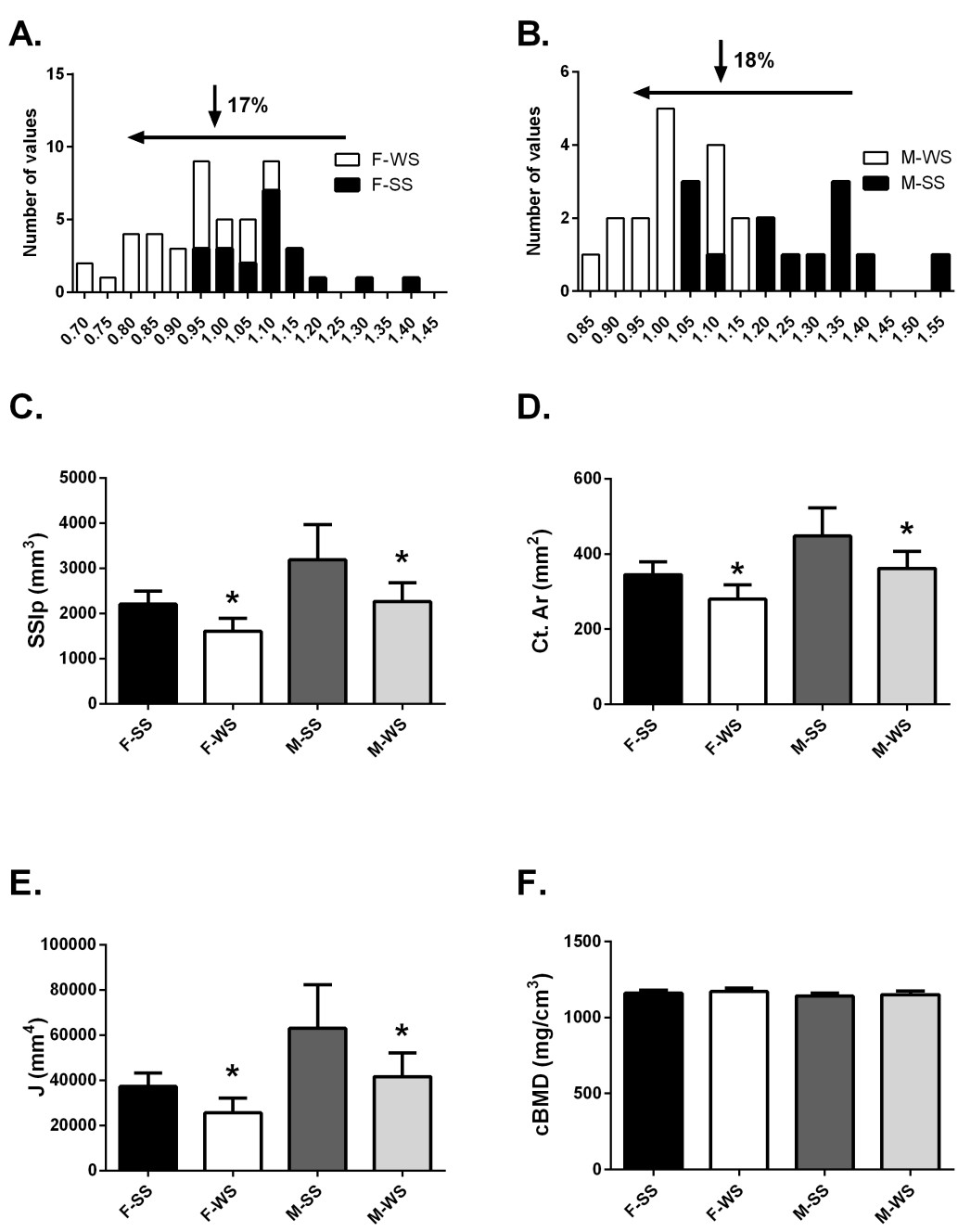

**Figure 2** **Comparison of the "weaker for size" (F-WS, M-WS) groups and "stronger for size" (F-SS, M-SS) groups.** Comparison of the "weaker for size" (F-WS, M-WS) groups and "stronger for size" (F-SS, M-SS) groups. (A) Robusticity for the WS group was 17% less in females compared to the SS group and (B) 18% less in males. (C) The WS groups for both females and males were significantly weaker than the SS groups ($p = 0.0001$). (D) The cortical area for both females and males in the WS groups were significantly smaller than the SS groups ($p = 0.0003$). (E) The largest difference between WS and SS groups was in the polar moment of inertia (J) ($p = 0.0003$). (F) No differences were found between groups for volumetric bone mineral density (cBMD).

**Table 1** Comparison of anthropometric and muscle strength and function variables between the SS group ("stronger for size") and the WS group ("weaker for size") in both Females and Males.

| | Females | | Males | |
|---|---|---|---|---|
| | F-SS | F-WS | M-SS | M-WS |
| **Anthropometrics** | | | | |
| Body weight (kg) | 65.0 (8.9) | 59.4 (8.9)[*] | 75.7 (9.0) | 73.4 (10.8) |
| Height (m) | 1.68 (0.08) | 1.63 (0.09) | 1.78 (0.10) | 1.75 (0.05) |
| Body fat % | 20.4 (6.1) | 21.0 (4.8) | 9.5 (3.7) | 13.5 (6.8) |
| Tibial Le (mm) | 356.1 (27.6) | 358.5 (32.1) | 386.3 (29.5) | 382.9 (31.0) |
| **Muscle** | | | | |
| Muscle area (mm$^2$) | 4,399 (649) | 4,492 (682) | 5,820 (710) | 5,191 (898) |
| Grip strength (N/kg) | 6.0 (1.0) | 6.0 (1.1) | 8.1 (1.2) | 7.4 (1.5) |
| 1 RM leg press/BW | 2.6 (0.7) | 2.3 (0.5) | 3.1 (1.0) | 3.1 (0.9) |
| Relative power (W/kg) | 50.8 (8.4) | 52.4 (7.4) | 60.8 (9.6) | 62.3 (10.8) |

**Notes.**
Values are presented as mean + SD.
[*]indicates difference from SS group $p < 0.05$.

**Table 2** Percentage of participants in the WS "weaker for size" group.

| | Females | Males |
|---|---|---|
| Track/cross country | 44% | 43% |
| Ball sports[*] | 30% | 33.3% |
| Swimming | 72% | N/A |
| Referent | 75% | 82% |

**Notes.**
[*]Included volleyball and soccer for females and soccer and basketball for males.

for individuals with larger body size (BW*Le) (Females $R^2 = 0.34$; Males $R^2 = 0.48$) (Fig. 3A). Cortical area (Ct.Ar) was also larger as the robusticity of the bone increased (Females $R^2 = 0.58$; Males $R^2 = 0.76$) (Fig. 3B). Robusticity correlated significantly with Ct.Ar for both females and males after accounting for body size (Bw*Le), the slopes and intercepts of these partial regression lines were not different. Slender bones (lower robusticity values) had less Ct.Ar than more robust tibias (Females $R^2 = 0.56$; Males $R^2 = 0.63$) (Fig. 3C). Cortical area adjusted for body size and regressed against robusticity was similar to the relationship reported in previous studies (*Jepsen, 2011*; *Jepsen et al., 2011*).

Six athletes in the "weaker for size" group were found to have positive residual values from the Ct.Ar vs BW*Le regression (Fig. 3A). These athletes are found in the top left quadrant of Fig. 3C, indicating that they had negative residuals for robusticity but a positive residual value for Ct.Ar after accounting for BW*Le. Slender bones had smaller cortical areas, but some athletes adequately adapted their tibia yet did not have the same functionality (bone strength/body size) as athletes with more robust bones.

## DISCUSSION

The current results for Division II collegiate athletes and referents were consistent with other studies of young healthy adults (*Jepsen, 2011*; *Jepsen et al., 2013*) indicating that a

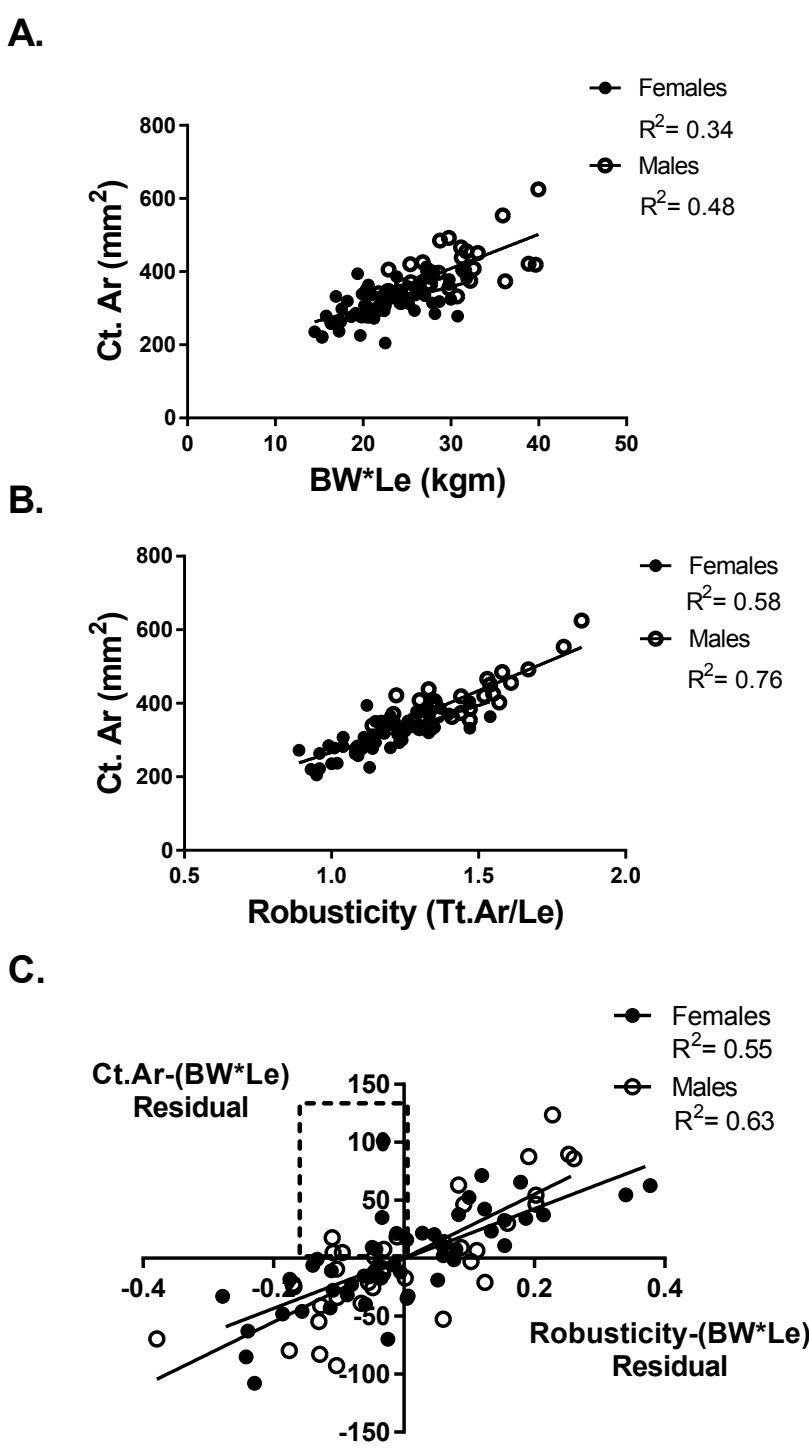

**Figure 3 Cortical area and robusticity.** (A) Cortical area (Ct.Ar) increased as the magnitude of loading (BW*Le) for both females and males. (B) Cortical area (Ct.Ar) was greater as the robusticity of the bone increased for both females and males. (C) Robusticity correlated significantly with Ct.Ar for both females and males after accounting for loading (Bw*Le). The slopes and intercepts of these regression lines were not different.

range of bone strength values based on robusticity occurs in athletes. Tibia robusticity varied ∼2 fold for both females and males in the current study similar to the range found in healthy young populations (*Jepsen et al., 2011*). Bone strength (SSIp) values ranged 83% (range/average) for females and 133% in males. The athletes did not all develop similar bone strength relative to body size and some athletes developed bones that were "weaker for size" (WS) and other athletes developed bones that were "stronger for size" (SS). A significant difference in average SSIp of approximately 27% was found between the tibias that were slender relative to BW*Le (WS group) compared to robust tibias (SS group). The "weaker for size" (WS) individuals also had significantly lower cortical area and polar moment of inertia (J) values compared to the SS group with no difference in cortical bone mineral density (cBMD). There were no differences in height or body fat percentage between groups however the athletes in the female SS group were heavier. In addition, no differences in muscle strength were found. The between groups variation in strength values is normal in healthy populations (*Jepsen et al., 2013*). However, athletes perform loading activities outside the norm and thus athletes with low robusticity in their tibia may have weaker bone strength and may be susceptible to repetitive loading injuries.

## Robusticity and bone strength

Athletes as a group typically have greater bone strength compared to control groups (*Heinonen et al., 2002*; *Kontulainen et al., 2003*; *Greene et al., 2012*; *Korhonen et al., 2012*; *Warden & Roosa, 2014*). Athletes presumably undergo bone functional adaptation to be able to withstand the loading demands of their sport without injury (*Ruff, Holt & Trinkaus, 2006*; *Hughes et al., 2016*). Bones functionally adapt to their loading as described by Wolff's Law (*Wolff, 1892*). The proposed stimulus for this adaptation is mechanical strain as described by the mechanostat theory introduced by Harold Frost (*Frost, 2003*). However, bone strength in the current study did vary based on bone robusticity (*Jepsen et al., 2011*; *Jepsen et al., 2013*; *Jepsen, Bigelow & Schlecht, 2015*) and the robusticity range in the athletes in the current study was similar to healthy populations (1.1 fold for females and males). The skeletal system is functional over a range of bone strengths in healthy populations and although slender bones tend to be less strong compared to robust bones (*Jepsen et al., 2013*) they are strong enough to withstand daily loads. A previous study reported a 2-fold difference in bone stiffness was not pathological but a natural variant that is expected in populations (*Jepsen et al., 2013*). However, slender bones have been associated with increased risk of stress fracture (*Beck et al., 2000*; *Popp et al., 2009*; *Schnackenburg et al., 2011*; *Jepsen et al., 2013*). Male military recruits presenting with a stress fracture had 5.3% lower robusticity and 11% lower tibial stiffness (*Jepsen et al., 2013*) compared to non-fracture recruits. In the current study, the "weaker for size" group had 27–29% lower SSIp values and approximately 17% lower robusticity values compared to the SS group. Decreased strength for size may not be a problem for activities of daily living in healthy populations but may become a problem under extreme loading conditions such as those experienced by athletes during training and competition.

Bone structure and architecture adapts as a response to mechanical loading with a goal to maintain optimum strain levels during the performance of activities. Therefore, if activity

levels increase, the architecture of the bone may change to maintain the strain levels (*Rubin & Lanyon, 1984*; *Forwood, 2008*; *Hughes et al., 2016*). However, if biological processes are unable to adapt bone traits such as cortical area, moment of inertia, and tissue mineral density to establish the same level of functionality between robust and more slender bones then strength will be affected. Athletes even after optimal training for their sport may have weaker bones relative to body size based on robusticity. Robusticity is established early in life by 2 years of age (*Pandey et al., 2009*). The variation in robusticity in healthy populations is in part based on genetics and as well as the activity levels of individuals. Skeletal robusticity has been used as an outcome measure to study mobility of groups in anthropological studies, the greater amounts of terrestrial locomotion have been linked to greater lower limb robusticity (*Carlson & Marchi, 2014*). Adolescence is a time period that elicits a bone adaptive response (*Kannus et al., 1995*; *Forwood, 2008*), athletes who start their sport during or prior to puberty have long term effects on bone structure (*Warden & Roosa, 2014*; *Jackowski et al., 2014*). Therefore, collegiate level athletes who are assumed to have started their athletic careers prior or during puberty would have the best chance to optimize their robusticity and bone strength. However, the coefficient of variation for robusticity in the current study was 15%, which was very similar to variations found in studies of healthy populations (*Pandey et al., 2009*). Even in trained athletes, robusticity may affect their ability to develop adequate bone strength for the demands of sport.

## Cortical area and bone strength relative to body size

Injury occurs when the loading on a tissue exceeds the strength of that tissue. For bone, if the loading from daily activities and/or sport and exercise exceed the bone strength then stress fractures may result. To avoid injury, athletes must either increase the strength of their bones or decrease the loading on their bones. Athletes ideally want to maximize bone strength while minimizing bone mass. Slender bones result in greater tissue strains potentially damaging the cortical matrix and increasing the probability of fracturing (*Burr et al., 1998*) and have been associated with stress fracture in military recruits (*Jepsen et al., 2013*). However, slender phenotypes are not indicative of a bone that is "poorly adapted". An individual can have lower tibia strength per body size and have less robust (slender) tibias for body size, and thus have reduced functionality, BUT have a well-adapted structure IF their cortical area is expected or greater than expected for body size (*Jepsen et al., 2013*). There may be a selective advantage in sport for a bone with minimal mass and maximal strength. While robust bones tend to be stronger they are also larger. Larger bones (increased mass) are metabolically expensive which may be a detriment for athletes in certain sports. The regression of robusticity and Ct.Ar after adjusting for body size (BW * Le) indicated that athletes and referents with less robust tibias had lower Ct.Ar in general (Figs. 3B, 3C). However, analysis of the data found some athletes with less robust tibias and greater than expected cortical area, these athletes had tibias that had impaired functionality but were optimally adapted based on their robusticity. These athletes had "weaker for size" bones but had adequate cortical area and therefore may have reached the limits in their ability to increase their bones' strength. These athletes may need to adjust their loading to reduce injury potential. A large percentage of swimmers were in the WS group, and

due to the lack of gravity during their sport this may be an advantage. Yet, the athletes in the "weaker for size" group that are involved in impact sports (soccer, volleyball and basketball) may have a greater risk for injury.

Tibia length, height, and percent body fat were all similar between the WS and SS groups for both females and males. There was also no difference in muscle function between the athletes that were "weaker for size" and those that were "stronger for size". The relative grip strength measured by hand dynamometer which is an indicator of total body strength was similar between groups. The relative leg extensor strength and lower body relative power were similar between groups. In addition, the muscle area measured at the 50% tibial site by pQCT was similar between groups. Athletes with similar anthropometry (body size) and body composition may have very different bone strength and adaptation capacities based on the robusticity of their bones.

### In-vivo bone strength measures

Although a direct measurement of bone strength is ideal, it is not feasible in studies using human subjects. Bone strength is determined by the size and shape (architecture) of a bone as well as the material properties of the bone (*Turner & Burr, 1993*; *Van der Meulen, Jepsen & Mikić, 2001*). Bone strength analysis via pQCT allows an analysis of bone strength, including both the structural and material properties. The parameter strength- strain index (SSIp), was developed to approximate bone strength in-vivo (*Ferretti et al., 2001*) and has been shown to be a good estimate of mechanical strength ex-vivo (*Augat et al., 1998*). Studies that measured strength via SSIp during development indicate a minimal change in cBMD but a large variation in structural variables of 300–400% (*Schoenau et al., 2001*). Males grow stronger bones due to the exclusive addition of bone mass on the periosteal surface where the effect on mechanical strength is much greater than adding bone mass to the endocortical surface. Women tend to add bone to the endocortical surface for future calcium needs during pregnancy and lactation (*Kovacs & Kronenberg, 1997*). In the current study, we found no differences in cBMD between the "weaker for size" and "stronger for size" groups; the main differences in bone strength stem from differences in bone cortical area and polar moment of inertia. Differences in cortical area are dependent on the robusticity of the bone that is not apparent by looking at muscle strength or body size of an athlete.

### Strengths and limitations

The limitations of our study include the cross-sectional nature of the data. We also did not track injury in our population due to the small participant number and the activity level of our reference group was not directly measured. In addition, other factors not measured in the current study may affect bone strength values in addition to body size and robusticity including systemic factors, nutrition and specific training load modalities. There were also advantages, as both our athlete and reference samples were ethnically diverse and previous studies have utilized more homogenous groups (*Jepsen et al., 2011*; *Jepsen et al., 2013*). Most of the athletes and referents in the current study were non-white (68.5%–90% dependent on group). Furthermore, previous studies suggest that a portion

of the variation in robusticity may be due to other aspects not represented by body size (BW*Le) including the type of activity, intensity, duration and age of onset of sport. Our population of collegiate athletes probably started their sport during adolescence (*Frisch et al., 1985*) and as a result was able to optimize their robusticity and functional adaptation. Yet the range of robusticity values of the athletes was similar to those found in healthy populations.

## CONCLUSIONS

Division II collegiate athletes had a variation in tibial robusticity and bone strength (measured by SSIp) similar to those previously reported in healthy populations (*Jepsen et al., 2011*; *Jepsen et al., 2013*; *Jepsen, Bigelow & Schlecht, 2015*). Athletes may tend to have stronger bones when viewed as a group but when analyzed as individuals bone strength was found to be dependent on robusticity. The athletes with slender bones were from all sports including track and field and ball sports but the majority were swimmers. Slender bones were constructed with less bone tissue and have less strength (SSIp) suggesting these bones were at a functional disadvantage compared to bones with higher robusticity (*Jepsen et al., 2011*). Slender bones may therefore be at a higher risk for fracture under extreme loading events but also yield benefits to some athletes (swimmers) due to their lower bone mass. Athletes with slender bones may have normal bone adaptation to loading based on their cortical area but still have bones that are functionally impaired. To avoid injury, robusticity of an athlete and the effect on bone strength and adaptation needs to be considered as training programs are designed. Finally, the athletes with slender bones were not easily identified by anthropometric or muscle strength variables.

## ACKNOWLEDGEMENTS

We would like to thank the California State University East Bay—Center for Student Research, Kinesiology Research Group (KRG) and the Department of Athletics.

### Funding

This work was supported by the Center for Student Research (CSR) California State University, East Bay. The funders had no role in study design, data collection and analysis, decision to publish, or preparation of the manuscript.

### Grant Disclosures

The following grant information was disclosed by the authors:
Center for Student Research (CSR) California State University, East Bay.

### Competing Interests

The authors declare there are no competing interests.

## Author Contributions

- Vanessa R Yingling conceived and designed the experiments, performed the experiments, analyzed the data, prepared figures and/or tables, authored or reviewed drafts of the paper, approved the final draft.
- Benjamin Ferrari-Church performed the experiments, analyzed the data, authored or reviewed drafts of the paper.
- Ariana Strickland performed the experiments, authored or reviewed drafts of the paper.

## Human Ethics

The following information was supplied relating to ethical approvals (i.e., approving body and any reference numbers):

The Institutional Review Board of California State University, East Bay granted ethical approval to carry out the study within its facilities (CSUEB-IRB-2014-004-F).

## Data Availability

The raw data are provided in Supplemental Information 1.

## Supplemental Information

Supplemental information for this article can be found online at http://dx.doi.org/10.7717/peerj.5550#supplemental-information.

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
