# Peer review of "Tibia functionality and Division II female and male collegiate athletes from multiple sports"

_PeerJ, doi:10.7717/peerj.5550_

## Round 0.1 · original submission · Major Revisions

I apologize for the delayed response but given the range of the first two reviewers I wanted to wait until I had a response from the third. As you can see, two reviewers suggested minor revisions, while Reviewer 2 suggested major changes. Comments from this reviewer were somewhat pessimistic that they could all be addressed, but I want to give you a chance to tackle these critiques and look forward to your revised manuscript.

All reviewers brought up important points. They and I are confused by the use of some of your terminology (i.e., functional inequivalence). In addition, while two reviewers provide praise on the clarity of the manuscript, I am sympathetic to the confusion expressed by Reviewer 2. I think the Results section could be much clearer. I also found multiple sections that were difficult to follow (lines 71-73, 272-282, and 287-290 are a few examples). As Reviewer 1 mentions there is a general lack of commas and odd use of punctuation. PeerJ does not utilize a copy editor during production, so these issues need to be address in revision prior to acceptance.

Given the variation seen in the results, a greater discussion of other influences on bone robusticity and strength would be beneficial. While strength appears to correlate with activity levels, are there other factors that change other than mechanical loading (i.e., Bertram and Swartz, 1991, Biol Rev 66:245-273 and more recent literature)?

Reviewer 1 ·

Basic reporting

The paper is very well written, but there are a few minor grammatical errors, specifically regarding the use (or lack of use) of commas, that make some lines difficult to read. For example, see lines 61-62 .

Experimental design

1. The term SSIp appears many times, but is never defined with an equation. A reference to previous development of the parameter is mentioned in the Discussion, but a definition should be in the text and appear earlier.

2. Were the regressions performed all significant? Some of the slopes reported are small (see Fig. 1B) and the correlations may not be statistically significant.

3. Lines 234-235: What statistical test was performed to test if the slopes were significantly different? ANCOVA?

Validity of the findings

The limitation of not knowing the length of time an athlete has played their sport, or the age in which they started the sport is a major one. However, it is appreciated that authors are aware of this and mention it in the Discussion. A larger sample size and analysis relative to sport would be a logical next step.

Reviewer 2 ·

Basic reporting

1. The terms used in this paper are very confusing. "Functionality" is in the title, so I wanted to know what was meant by that. Line 71 tells the reader that "The variability of robustness is associated with functional inequivalence" but inquivalence between what two things is not specified and no reference is given. Line 73 seems to equate "bone size" (or possibly variation in bone size) with "robusticity" (are robustness and robusticity the same thing?). Line 74 seems to equate "strength" with "function". Line 75 seemes to indicate that "slender" bones have "low robusticity". On lines 82-83, it says that it is important to know if "bone strength is affected by robusticity (functional inequivalence)". Lines 195-197 state "Bone mechanical functionality is traditionally assessed by plotting a morphological bone trait (SSIp) against a measure of bone size." Having clear definitions for all these terms would help the reader to understand the stated purpoise of the study that comes at the end of the Introduction.

2. The hypothesis doesn't make sense. Lines 102-104 state "It was hypothesized that there would be a difference in bone strength and cortical area dependent on robusticity in Division II collegiate athletes." One difference, or are two differences being hypothesized? Is there a hypothesized difference in bone strength (or area) between two groups or conditions? Or is it the case that a corrlation between strength (or area) and robusticity is being hypothesized?

3. The paragraph beginning on line 85 doesn't have a clear topic. It starts off with a topic sentence about the effects of high-impact loading, then discusses variability in bone strength, then states the purpose of the study. I suggest revising this paragraph to give it a clearer first sentence about variability and put the purpose and hypotheses in a separate paragraph.

4. The title needs attention. Should the "and" be "in"?

Experimental design

1. How can we make comparisons between male and female athletes when these athletes played different sports?

2. When muscle strength and power measures are described in the Methods (starting line 127) it is not at all clear why these measures are relevant to the stated purpose of the study.

Validity of the findings

1. I suggest rewriting your Results so that each paragraph begins with a clear statement of a key result. Also, why is the formation of groups being described in the Results (line 240) and not in the Methods? What your results suggest (line 269) and comparisons to the work of others (line 299) should be in the Discussion, not the Results.

Additional comments

71-73: This sentence does not make sense - is there supposed to be a comma after "inequivalence"?

110: University students?

114: What is a "referent"? I am not used to seeing this term in this context.

114: Why do some participants have "unknown" race or ethnic background? How was this determined?

132: What is "Hammer Strength"?

143: What is a "Vertec"? I know what this is, but you need to tell the reader.

153: Is "Absolute Peak Power" the same as "Peak Power" from line 151?

157: "dominant-side tibia"?

162: Is this the medial epicondyle of the tibia or the femur? If it is the tibia, why is knee angle important?

168: "were"

184: What assurance do you have that the variables did not depend on which did the analysis? Do the reported CVs address this, or just the within-technician precision?

331: Typo - "Wolff"

Reviewer 3 ·

Basic reporting

This manuscript is clearly written, well-structured, and references the most up-to-date studies that are relevant to the study. However, the 4% diaphyseal site was mentioned, outlined in the methods, but not reported. This would clearly be a welcomed addition to the data, as it provides a trabecular assessment, however it may be too much to include within this manuscript. Therefore, I suggest removing mention of this site from the paper. Also, two terms are used that are contentious and/or not well understand that you may want to consider changing; functional inequivalence and optimal/optimized/optimization.
Functional inequivalence is not well understand in the bone field and leads to much confusion among those unfamiliar with the term. You essentially could remove this term and not lose anything as you already state 'natural variation relative to robustness.' Optimal is very contentious in the bone field as it implies that under normal loading conditions one bone is better than another, when in fact, any bone that maintains its integrity is sufficient irrespective of any morphological/compositional differences. For this reason I would use a term such as 'adequate.'

Experimental design

This article meets all the criteria of an original, well-designed study. I have no comments.

Validity of the findings

Overall the findings are consistent with prior studies by others, and demonstrates that the concept of robustness holds up in a more extreme loading setting. Obviously it would be nice to know how these bones actually performed with some clinical measure of stress fracture incidences but this manuscript can stand alone, with hopefully a future study including clinical outcomes.

Additional comments

Overall this is a very nice study that expands prior work conducted using military recruits and cadaveric specimens. The authors provided adequate background, detailed methodologies, and did not over-interpret their results. Their findings are consistent with previously published work and is a welcomed contribution to the bone literature.

---

## Round 0.2 · Minor Revisions

Thank you for your revised manuscript and detailed explanation of the changes in response to the Reviewers' critiques. It is much improved and much more clear. This paper is nearly ready for acceptance, however I have identified a number of typos and grammatical issues (particularly pertaining to commas). See the attached pdf. This should not be considered an exhaustive list. As the manuscript goes directly to production upon acceptance, I urge you to thoroughly review the manuscript at this time.

---

## Round 0.3 · Minor Revisions

A very cursory review of the tracked changes manuscript showed that:

Line 151 Vertec is not capitalized
Line 268 Although is misspelled
Line 461 had an extra period.

As the article is otherwise Acceptable and as PeerJ does not enlist copy editing service it is best to submit a manuscript as free of errors as possible. This is not an exhaustive list, so please review your manuscript carefully.

---

## Round 0.4 · accepted · Accept

Thank you for your last revisions on the manuscript. I now have marked it as Accepted. I hope you view that the review process has been beneficial.